# Survival Differences by Comorbidity Burden among Patients with Stage I/II Non-Small-Cell Lung Cancer after Thoracoscopic Resection

**DOI:** 10.3390/cancers15072075

**Published:** 2023-03-30

**Authors:** Meghann Wheeler, Shama D. Karanth, Hiren J. Mehta, Danting Yang, Livingstone Aduse-Poku, Caretia Washington, Young-Rock Hong, Dongyu Zhang, Michael K. Gould, Dejana Braithwaite

**Affiliations:** 1Department of Epidemiology, University of Florida College of Public Health and Health Professions, Gainesville, FL 32603, USA; 2University of Florida Health Cancer Center, Gainesville, FL 32603, USA; 3Aging & Geriatric Research, Institute on Aging, University of Florida, Gainesville, FL 32603, USA; 4Department of Medicine, Division of Pulmonary and Critical Care Medicine, University of Florida, Gainesville, FL 32603, USA; 5Department of Health Services Research, Management & Policy, University of Florida, Gainesville, FL 32603, USA; 6Medical Device Epidemiology and Real-World Data Science, Johnson & Johnson, New Brunswick, NJ 08933, USA; 7Department of Health Systems Science, Kaiser Permanente Bernard J. Tyson School of Medicine, Pasadena, CA 91107, USA; 8Department of Surgery, University of Florida, Gainesville, FL 32603, USA

**Keywords:** lung cancer, survival, comorbidity, minimally invasive surgery, epidemiology

## Abstract

**Simple Summary:**

There is limited research on the relationship between comorbidity burden and survival among patients with stage I/II non-small-cell lung cancer (NSCLC). Thus, the purpose of this study was to compare survival by comorbidity burden among stage I/II NSCLC patients who have received thoracoscopic surgery as their primary treatment. We found that increasing comorbidity burden was associated with a higher risk of all-cause mortality and that the impact of comorbidity on survival was stronger in female patients with NSCLC than in male patients. These findings highlight the importance of considering comorbidities to optimize the selection of candidates for thoracoscopic resection.

**Abstract:**

We sought to compare overall survival (OS) by comorbidity burden among patients with stage I/II non-small cell lung cancer (NSCLC) who received thoracoscopic resection. Utilizing data from the National Cancer Database, we conducted a survival analysis among patients aged 50+ with stage I/II NSCLC who received thoracoscopic resection between 2010 and 2017. The comorbidity burden was measured by the Charlson comorbidity index (CCI, 0, 1, 2+). Multivariable Cox proportional hazard models were used to compare overall survival relative to the CCI (CCI of 0 as the referent). Subgroup analyses were conducted considering sex, age groups, days from diagnosis to surgery, facility type, laterality, and type of surgery. For this study, 61,760 patients were included, with a mean age of 69.1 years (SD: 8.5). Notably, 51.2% had a CCI of 0, 31.8% had a CCI of 1, and 17.0% had a CCI of 2+. Most participants were non-Hispanic White (87.5%), and 56.9% were female. We found that an increase in the CCI was associated with a higher risk of all-cause mortality (CCI 1 vs. 0 aHR: 1.24, 95% CI: 1.20–1.28; CCI 2+ vs. 0 aHR: 1.51, 95% CI: 1.45–1.57; p-trend < 0.01). Our subgroup analysis according to sex suggested that the association between CCI and risk of death was stronger in women.

## 1. Introduction

Lung cancer accounts for the most cancer-related deaths in the US [1]. In 2022, an estimated 236,740 people in the US were diagnosed with lung cancer, and lung cancer deaths accounted for 21% of all cancer-related deaths [1]. With the dissemination of lung cancer screening, the proportion of patients diagnosed with stage I/II lung cancer has continuously increased in the US. Between 2013 and 2017, the average annual percent change in early-stage lung cancer cases was 6.88%, while the average annual percent change in lung cancer cases diagnosed at an advanced stage was −2.74% [2]. Compared with stage III/IV lung cancer patients, patients with stage I/II disease have a more favorable prognosis after treatment. 

Non-small-cell lung cancer (NSCLC) accounts for ~85% of incident lung cancer cases in the US. In clinical practice, surgical resection is usually the first choice when treating early-stage NSCLC [3]. In recent years, thoracoscopic resection has become increasingly favored over open surgical resection, as thoracoscopy is associated with fewer postprocedural complications, shorter length of hospital stay, and faster recovery [3,4,5,6,7]. Some clinical studies suggest that patients who undergo thoracoscopic resections have more favorable survival outcomes than those undergoing open surgeries [8,9,10,11,12,13].

Lung cancer is an age-related disease, with a median age at diagnosis of 71 years in the US according to data from the Surveillance, Epidemiology, and End Results (SEER) Program [14]. Therefore, patients with lung cancer may have a higher risk of co-existing illnesses than those with other types of cancer, who tend to be diagnosed at younger ages. Clinical evidence suggests that comorbidity burden is an independent prognostic factor among stage I/II NSCLC patients undergoing open surgery [15]. However, limited studies have explored how the risk of death among stage I/II NSCLC patients treated with thoracoscopic resection varies by comorbidity burden. Understanding the impact of comorbidity burden on the survival of these patients receiving thoracoscopic resection can be fundamental for decision making in clinical practice for stage I/II NSCLC.

Therefore, we leveraged the National Cancer Database (NCDB) to compare the differences in overall survival (OS) by comorbidity burden among adults with early-stage NSCLC who underwent thoracoscopic resection.

## 2. Methods

### 2.1. Data Source and Patient Selection

The NCDB is a joint project of the American Cancer Society and the Commission on Cancer of the American College of Surgeons. The NCDB is a hospital-based cancer registry that captures 72% of all newly diagnosed cancers in the US [16]. The data are extracted from electronic health records (EHRs) by certified tumor registrars. Healthcare facilities that participate in the NCDB are also required to capture information regarding their patients’ cancer care that occurred outside of their facilities, including care from facilities that are not accredited by the Commission on Cancer. All data are validated before being released [17], and the variables collected include patient demographic information, tumor characteristics, first-course cancer treatment, and survival [18]. We obtained NCDB data for patients diagnosed with NSCLC between 2010 and 2017 and selected our analytic sample using the following criteria: (1) patients with primary stage I/II NSCLC at the time of diagnosis; (2) patients aged 50 or greater at the time of diagnosis, as there are relatively few younger patients with a high comorbidity burden; (3) patients who received thoracoscopic resection; and (4) those with no missing data on the Charlson comorbidity index (CCI) score, mortality, or our selected covariates. This yielded a total of 61,760 participants for analysis. A flowchart of participant selection is presented in Figure 1. This study was exempt from IRB review as the data used in this study were obtained from a de-identified NCDB file.

### 2.2. Exposure, Outcome, and Covariates

The exposure of interest in this analysis was comorbidity burden, measured using the Charlson–Deyo version of the CCI and categorized as an ordinal variable [19]. Because only a small fraction of stage I/II NSCLC cases had a CCI score higher than 2, the CCI was truncated to 0, 1, and 2+ in our study, with a score of 0 indicating no comorbid conditions. The outcome of interest in this analysis was all-cause mortality, with the follow-up time measured as time from diagnosis to death or last contact, whichever occurred first. Facilities participating in the NCDB are expected to report patient follow-up annually and have a follow-up rate of at least 90% for all patients diagnosed within the prior 5 years [17]. Patient-level characteristics included age at diagnosis, sex (male or female), and race/ethnicity (Non-Hispanic (NH) White, NH Black, American Indian, Asian/Pacific Islander, Hispanic, or other). Area-level covariates, which included education and income, were inferred by linking patients’ zip codes to the US Census data. Specifically, education was defined as the percentage of adults 25 years or older that did not graduate high school, and we used the median household income to reflect area-level income; in our study, these variables were derived using data from the American Community Survey and were categorized into equally proportioned quartiles based on all US zip codes (education: ≥17.6%, 10.9–17.5%, 6.3–10.8%, <6.3%; income: ≥USD 63,333; USD 50,354–USD 63,332; USD 40,227–USD 50,353; <USD 40,227). Provider-related covariates included insurance status (Medicare, Medicaid, private, uninsured, or other) and facility type (non-academic or academic). Histological subtype, stage, and laterality were the cancer-specific characteristics included in our study. Tumor histological subtype was determined by using the International Classification of Diseases for Oncology (ICD-O-3) morphology codes (adenocarcinoma, squamous cell carcinoma, large cell/neuroendocrine carcinoma, or other). Cancer stage was defined by the 6th or 7th edition of the American Joint Committee of Cancer TNM staging system, as they were the editions in use during the study period. Laterality was included as a cancer-related covariate, as it has been suggested that laterality may affect postsurgical survival in lung cancer [20]. Treatment-related factors included receipt of adjuvant chemotherapy, surgical margin status (no residual tumor, residual tumor not specified, microscopic residuals, macroscopic residuals, or indeterminate), days from diagnosis to surgery, and type of surgery (sublobar resection, lobectomy/bilobectomy, pneumonectomy, or other). The type of surgery was categorized based on prior literature [21]. Covariates were selected based on a priori knowledge regarding their associations with our exposure and outcome of interest as well as based on prior NCDB analyses [22,23].

### 2.3. Statistical Analysis

We first descriptively summarized study characteristics in the overall study population and according to CCI scores (0, 1, and 2+) by reporting the number of observations and percentages. Kaplan–Meier curves were used to visualize the probability of survival relative to CCI categories, and the log-rank test was used to examine if the risk of mortality differed relative to the CCI. A Kaplan-Meier curve including patients with missing values for the relevant covariates was also used to evaluate if data missingness affected the survival probabilities. Number of deaths and mortality rates were summarized for the overall study population and according to CCI categories. Age-adjusted and multivariable Cox proportional hazard models were used to examine the association between comorbidity burden (reference: CCI = 0) and risk of all-cause mortality. Two multivariable Cox models were conducted; therefore, hazard ratios (HRs) and 95% confidence intervals (95% CIs) were the measures of association in our study; the first model was adjusted for age, race/ethnicity, education, income, insurance, and facility type. The second model was additionally adjusted for the stage at diagnosis, histological type, days from diagnosis to surgery, and receipt of adjuvant chemotherapy. We assessed the proportional hazards assumption by checking Schoenfeld residuals [24]. In both multivariable Cox models, age and sex violated the proportional hazards assumptions; thus, we stratified the multivariable models by sex and included an interaction term between age and follow-up, and no violation was observed afterward. A trend test was performed by treating the CCI as a continuous variable in the model. Subgroup analyses were conducted considering sex (female vs. male), age groups (50–64, 65–74, 75+), days from diagnosis to surgery (<30 days vs. ≥30 days), facility type (academic vs. non-academic), laterality (left vs. right), and type of surgery (sublobar resection, lobectomy/bilobectomy, pneumonectomy, and other). In each subgroup analysis, we created an interaction term between the CCI and the variable used for stratification (i.e., sex, age group, days from diagnosis to surgery, etc.); the Wald test was used to assess if the interaction was significant, with a *p*-value < 0.05 suggesting statistical heterogeneity between subgroups. Four sets of sensitivity analyses were conducted. The first sensitivity analysis examined if the association obtained in the primary Cox model changed substantially when only analyzing participants who did not receive adjuvant chemotherapy. Then, to assess the stability of the HR relative to the CCI, we created an indicator variable of exclusion due to missing data in our selected covariates and included this indicator in the multivariable models with different sample sizes; in addition to the indicator of exclusion, all these models were adjusted for demographic variables (age, sex, and race/ethnicity) plus (1) socioeconomic variables (education and income), (2) provider-related factors (insurance and facility type), or (3) cancer-related characteristics (histological type, stage, use of chemotherapy, and days from diagnosis to surgery). For our third sensitivity analysis, we evaluated the proportions of deaths within 90 days after surgery relative to the CCI. In our final sensitivity analysis, we explored the relationship between the type of sublobar resection (wedge resection, segmental resection, or other), CCI, and mortality. All analyses were performed in SAS 9.4 (Cary, NC, USA) and R Studio V 4.1.1. All statistical tests were two-sided, and a *p*-value < 0.05 indicated statistical significance.

## 3. Results

A total of 61,760 patients who received thoracoscopy for treatment of stage I/II NSCLC between 2010 and 2017 were included in the present analysis. Of those, 51.2% had a CCI of 0, 31.8% had a CCI of 1, and 17.0% had a CCI of 2 or greater. The mean age at diagnosis was 69.1 years (SD: 8.5 years), approximately 57% were female, and a majority (87.5%) of the study population self-identified as NH White. Most patients had stage I NSCLC (88.5%), and the most common histological subtype was adenocarcinoma (53.9%). The majority of the patients had complete resection with no residual margins (95.6%). Among the 4.4% who did not receive complete resection, over 60% were treated in non-academic facilities, which may have a lower volume of surgical resections. The detailed distributions of the study characteristics are presented in Table 1. 

In our study, the median follow-up time of all patients was 39.2 months (IQR: 24.2–60.8). During the follow-up, 18,833 (30.5%) patients died. The Kaplan–Meier survival curve (Figure 2) relative to the CCI indicated that patients with higher CCI scores had a higher risk of all-cause mortality (log-rank *p* < 0.01). The Kaplan-Meier survival curve including patients with missing values for relevant covariates (Appendix A) produced similar results (log-rank *p* < 0.01). The all-cause mortality rate was 83.6 (95% CI: 82.5–84.8) per 1000 person-years. Among patients with a CCI of 0, the all-cause mortality rate was 68.7 (95% CI: 67.3–70.2) per 1000 person-years, and it increased with the CCI score (CCI = 1: 91.2 (95% CI: 89.1–93.3); CCI = 2+: 118.2 (95% CI: 114.8–121.7) per 1000 person-years). Similarly, the results of the multivariable Cox proportional hazard models (Table 2) showed that, compared with patients with CCI scores of 0, patients with CCI scores of 1 and 2+ had a 24% and 51% relative increase in the risk of all-cause mortality, respectively (full model aHR (CCI 1 vs. 0): 1.24, 95% CI: 1.20–1.28; aHR (CCI 2+ vs. 0): 1.51; 95% CI: 1.45–1.57; *p*-trend < 0.01).

Our subgroup analysis (Table 3) considering sex found that, although the associations between the CCI and all-cause mortality were positive and significant in both males and females, the magnitude of association was stronger for females when comparing CCI = 1 versus CCI = 0 (female aHR for CCI 1: 1.31, 95% CI: 1.25–1.37; male aHR for CCI 1: 1.17, 95% CI: 1.12–1.22; *p*-interaction < 0.01). This pattern was also observed among patients with a CCI of 2 or greater (female aHR: 1.65, 95% CI: 1.56–1.75; male aHR: 1.40, 95% CI: 1.33–1.47). We also found that the association between the CCI and all-cause mortality varied by days from diagnosis to surgery (*p*-interaction < 0.01), facility type (*p*-interaction < 0.01), and the type of surgery (*p*-interaction < 0.01); however, because of the overlap in 95% CIs of HR in these subgroups, their magnitude of heterogeneity was not as strong as heterogeneity by sex. Our subgroup analyses considering age and tumor laterality did not reveal significant interaction (Table 3). In our sensitivity analysis that only included patients who did not receive adjuvant chemotherapy, the effect measures for the CCI were not substantially different from those obtained in the main analysis (Appendix A). In models adjusting for the indicators of exclusion, the effect measures of CCI scores did not substantially change (Appendix A). Our sensitivity analysis for 90-day mortality revealed a similar trend to that of long-term mortality: Participants with a CCI of 0 had the lowest proportion of 90-day mortality (1.8%, 95% CI: 1.7–2.0%), and participants with a CCI of 2 or greater had the highest proportion of 90-day mortality (3.2%, 95% CI: 2.9–3.6%) (Appendix A). Finally, in our sensitivity analysis among patients who underwent sublobar resection only, effect measures were similar in terms of the type of sublobar resection and did not differ substantially from those obtained in the main analysis (Appendix A). 

## 4. Discussion

In the present analysis, we found that a high comorbidity burden was associated with lower overall survival among stage I/II NSCLC patients undergoing thoracoscopic resection. Our data indicated that the impact of comorbidity burden was stronger in females than in males, and we have a hypothesis to explain this phenomenon. Prior studies suggest that female cancer patients tend to have higher levels of frailty than male cancer patients [25,26,27]. For example, our prior study of data from the National Health and Nutrition Examination Survey showed that women with a prior cancer diagnosis were 15% more likely to have frailty than their male counterparts [27]. Therefore, it is possible that pre-existing frailty combined with the comorbidity burden could contribute to poorer survival outcomes for female patients [28].

To our knowledge, this is the first analysis leveraging national cancer registry data to explore the relationship between comorbidity burden and survival among patients undergoing thoracoscopic resection for treatment of early-stage NSCLC. Prior studies have shown that lung cancer patients with comorbidities have poor short-term outcomes following minimally invasive surgery (MIS); specifically, preoperative comorbidity is associated with an increased risk of complications, longer hospital stays, and increased risk of in-hospital mortality [29]. While there is limited research exploring comorbid conditions in relation to long-term outcomes after MIS such as thoracoscopic resection, our findings are similar to what has been reported among patients receiving open surgical resection for the treatment of NSCLC. For instance, an analysis of surgically resected NSCLC patients (N = 3152) within the Danish Lung Cancer Registry (DLCR) found that increased CCI was associated with decreased 5-year survival in early-stage NSCLC patients (for pT1 NSCLC, high comorbidity: 38%, low comorbidity: 69%) [30].

Several underlying mechanisms may explain our findings from this study. Cellular senescence, a hallmark of aging, is the condition in which a cell undergoes permanent growth arrest and is no longer able to proliferate, [31] and it plays an important role in the development of age-related diseases (e.g., cardiovascular disease, dementia, diabetes, etc.) [31,32,33]. Therefore, patients with a high burden of comorbidities are more likely to have cellular senescence, which can adversely affect survival following cancer treatment. In addition, senescence in immune cells can induce immunosenescence, compromising the immune surveillance of NSCLC patients and increasing the risk of cancer-related adverse events such as cancer recurrence or cancer-specific death [34,35,36]. In addition, chronic inflammation, which is tightly connected to the pathogenesis of many chronic diseases [37,38,39,40,41], can negatively impact postsurgical outcomes and cancer prognosis; this can also partially explain why NSCLC patients with a higher burden of comorbidities have shorter survival. Further, NSCLC patients—particularly those with pre-existing chronic conditions—are often at risk of cardiac and/or respiratory complications, including but not limited to pneumonia, atrial fibrillation, and venous thromboembolism [42,43]. Such complications may occur shortly after surgical resection and often cause permanent damage, serving as competing risks for mortality. NSCLC patients with postoperative complications have been found to have reduced 5-year overall survival compared with their counterparts who did not develop postoperative complications [42]. Other factors that can influence prognosis following the surgical resection of NSCLC among patients with an increased comorbidity burden include smoking history [44]—which can also influence the development of postsurgical complications—and histological type at diagnosis [45]. Finally, it is worth noting that the factors related to healthcare delivery may also play a role in the increased risk of mortality seen in patients with higher CCI. Cancer patients with a high comorbidity burden face significantly higher healthcare costs, they are more likely to receive conservative adjuvant therapy, and coordination of healthcare delivery is more complex for patients with multimorbidity, all of which can increase the risk of mortality [46,47,48]. 

Several limitations should be noted when interpreting our results. First, the NCDB does not provide information on the cause of death, leaving us unable to analyze deaths due to cancer versus deaths due to other causes. Second, residual confounding may exist in our analysis. For example, smoking is an important factor that should be considered in the survival analysis of lung cancer patients; however, the NCDB does not collect this information. Third, differences in staging classification over the years included in the present analysis may also lead to potential stage shifting. The use of both immunotherapies and targeted therapies in adjuvant and neoadjuvant settings may also influence the obtained mortality estimates. Finally, a majority of the patients included in our analysis were non-Hispanic White, indicating that the generalizability of our findings may be compromised in minority populations. Despite these limitations, our analysis has notable strengths. The NCDB is one of the largest cancer registries in the world [17], and it collects standardized, high-quality data that undergo strict quality control methods, which enables the validity of measurement and robust power in analysis. Furthermore, our sample included over 60,000 NSCLC cases; this large sample size strengthens the precision of our estimates and is significantly larger than many studies of NSCLC patients. Finally, our subgroup analyses enabled us to explore the potential effect modification induced by covariates.

## 5. Conclusions

Our study found that a high comorbidity burden was associated with increased mortality among stage I/II NSCLC patients undergoing thoracoscopic resection, even after controlling for clinically important patient characteristics. This finding indicates that comorbidity burden is an independent predictor of NSCLC patients’ survival following thoracoscopic resection, highlighting the need for clinicians to consider comorbidity burden when evaluating whether a patient is a suitable candidate for thoracoscopy, and to consider comorbidity burden when discussing the benefits and harms of thoracoscopy with their patients. Further research, particularly with prospective study design, is necessary to confirm that comorbidity burden has a more substantial impact on the survival of female stage I/II NSCLC patients undergoing thoracoscopy and to explore the potential biological mechanisms that influence survival differences relative to sex among NSCLC patients.

## Figures and Tables

**Figure 1 cancers-15-02075-f001:**
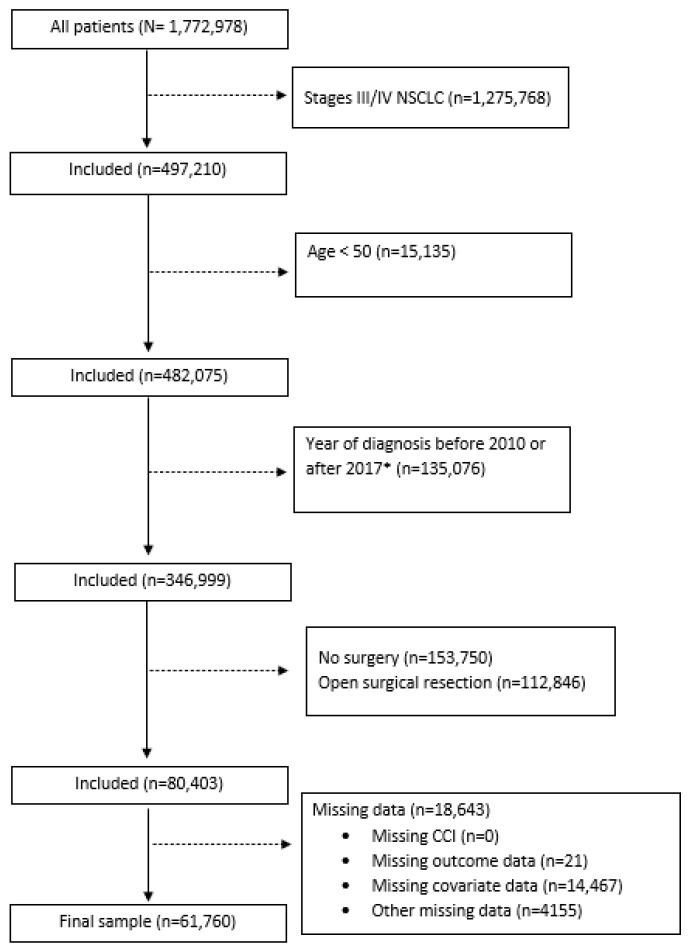
Flowchart for study participants selection. Abbreviations: NSCLC: non-small-cell lung cancer; CCI: Charlson comorbidity index. *Diagnosis years 2010–2017 selected due to inconsistencies in surgical resection classification for cases diagnosed prior to 2010 and lack of survival data for cases diagnosed in 2018.

**Figure 2 cancers-15-02075-f002:**
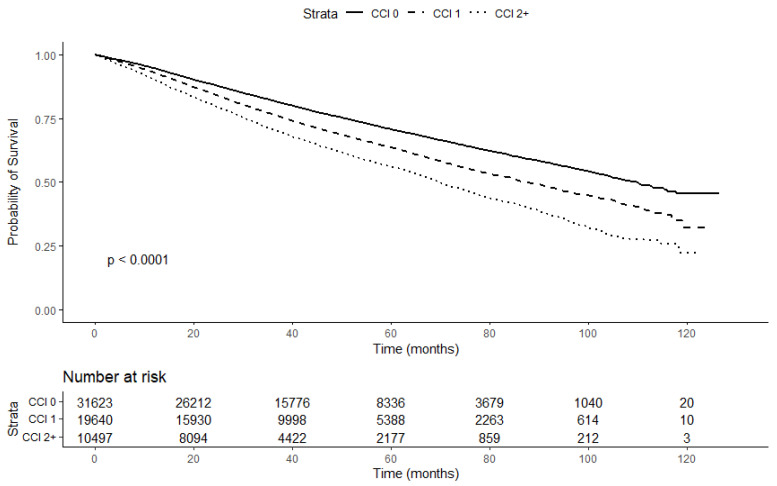
Kaplan–Meier survival curve stratified by CCI.

**Table 1 cancers-15-02075-t001:** Summary of study characteristics.

Characteristic	All (N = 61,760)	CCI 0 (N = 31,623)	CCI 1 (N = 19,640)	CCI 2+ (N = 10,497)
	No. (%)	No. (%)	No. (%)	No. (%)
**Age, mean (SD), y**	69.1 (8.5)	68.8 (8.8)	69.0 (8.4)	70.0 (8.1)
**Age group**				
50–64	18,327 (29.7)	9994 (31.6)	5753 (29.3)	2580 (24.6)
65–74	26,088 (42.2)	12,829 (40.6)	8534 (43.5)	4725 (45.0)
≥75	17,345 (28.1)	8800 (27.8)	5353 (27.3)	3192 (30.4)
**Sex**				
Male	26,612 (43.1)	12,630 (39.9)	8687 (44.2)	5295 (50.4)
Female	35,148 (56.9)	18,993 (60.1)	10,953 (55.8)	5202 (49.6)
**Race/Ethnicity**				
NH White	54,043 (87.5)	27,634 (87.4)	17,299 (88.1)	9110 (86.8)
NH Black	4787 (7.8)	2314 (7.3)	1508 (7.7)	965 (9.2)
American Indian	141 (0.2)	61 (0.2)	43 (0.2)	37 (0.4)
Asian/Pacific Islander	573 (0.9)	415 (1.3)	119 (0.6)	39 (0.4)
Hispanic	2106 (3.4)	1140 (3.6)	636 (3.2)	330 (3.1)
Other	110 (0.2)	59 (0.2)	35 (0.2)	16 (0.2)
**Income ^#^**				
≥USD 63,333	25,163 (40.7)	13,946 (44.1)	7492 (38.2)	3725 (35.5)
USD 50,354–USD 63,332	14,473 (23.4)	7304 (23.1)	4601 (23.4)	2569 (24.5)
USD 40,227–USD 50,353	12,444 (20.2)	5935 (18.8)	4179 (21.3)	2330 (22.2)
<USD 40,227	9680 (15.7)	4438 (14.0)	3368 (17.2)	1874 (17.9)
Insurance				
Medicare	41,242 (66.8)	20,141 (63.7)	13,350 (68.0)	7751 (73.8)
Medicaid	2606 (4.2)	1213 (3.8)	917 (4.7)	476 (4.5)
Private	16,603 (26.9)	9627 (30.4)	4901 (25.0)	2075 (19.8)
Not Insured	628 (1.0)	309 (1.0)	238 (1.2)	81 (0.8)
Other	681 (1.1)	333 (1.1)	234 (1.2)	114 (1.1)
**Facility Type**				
Academic facility	27,339 (44.3)	14,653 (46.3)	8271 (42.1)	4415 (42.1)
Non-academic facility	34,421 (55.7)	16,970 (53.7)	11,369 (57.9)	6082 (57.9)
**Histology**				
Adenocarcinoma	33,260 (53.9)	18,003 (53.9)	10,241 (52.1)	5016 (47.8)
Squamous cell	12,435 (20.1)	5127 (16.2)	4437 (22.6)	2871 (27.4)
Large cell	307 (0.5)	138 (0.4)	110 (0.6)	59 (0.6)
Other ^¥^	15,758 (25.5)	8355 (26.4)	4852 (24.7)	2551 (24.3)
**TNM Stage**				
I	54,674 (88.5)	28,027 (88.6)	17,362 (88.4)	9285 (88.5)
II	7086 (11.5)	3596 (11.4)	2278 (11.6)	1212 (11.6)
**Adjuvant Chemotherapy**				
Yes	9677 (15.7)	5043 (16.0)	3107 (15.8)	1527 (14.6)
No	52,083 (84.3)	26,580 (84.1)	16,533 (84.2)	8970 (85.5)
**Days from Diagnosis to Surgery**				
0–29	31,174 (50.5)	16,551 (52.3)	9678 (49.3)	4945 (47.1)
30–59	18,006 (29.2)	9050 (28.6)	5846 (29.8)	3110 (29.6)
60–89	7313 (11.8)	3493 (11.1)	2380 (12.1)	1440 (13.7)
≥90	5267 (8.5)	2529 (8.0)	1736 (8.8)	1002 (9.6)
**Surgical Margins**				
No residual tumor	59,033 (95.6)	30,371 (96.0)	18,695 (95.2)	9967 (95.0)
Residual tumor, NOS	774 (1.3)	365 (1.2)	269 (1.4)	140 (1.3)
Microscopic residuals	1082 (1.8)	482 (1.5)	392 (2.0)	208 (2.0)
Macroscopic residuals	116 (0.2)	45 (0.1)	41 (0.2)	30 (0.3)
Indeterminate	755 (1.2)	360 (1.1)	243 (1.2)	152 (1.5)
**Extent of Resection**				
Sublobar resection	18,007 (29.2)	8580 (27.2)	5996 (30.5)	3431 (32.7)
Lobectomy/bilobectomy	42,544 (68.9)	22,421 (70.9)	13,277 (67.6)	6846 (65.2)
Pneumonectomy	455 (0.7)	264 (0.8)	123 (0.6)	68 (0.7)
Other	754 (1.2)	358 (1.1)	244 (1.2)	152 (1.5)
**Laterality**				
Right	36,374 (58.9)	18,642 (59.0)	11,632 (59.2)	6100 (58.1)
Left	25,131 (40.7)	12,853 (40.6)	7931 (40.4)	4347 (41.4)
Other	255 (0.4)	128 (0.4)	77 (0.4)	50 (0.5)

Abbreviations: SD: standard deviation; CCI: Charlson comorbidity index; NH: non-Hispanic; NOS: not specified. Column percent is reported in the table. ^#^ Median household income, estimated by matching the zip code of the patient recorded at the time of diagnosis and adjusted for 2016 inflation. The 2016 poverty guidelines define poverty as a household income of USD 16,020 or below in a two-person household. ^¥^ Includes the following ICD-O-3 codes: 8013, 8014, 8020, 8022, 8030, 8031, 8032, 8033, 8046, 8052, 8071, 8072, 8073, 8074, 8075, 8084, 8141, 8144, 8154, 8200, 8230, 8240, 8241, 8243, 8244, 8245, 8249, 8251, 8252, 8253, 8254, 8255, 8262, 8341, 8520, 8551, 8560, 8562, and 8575.

**Table 2 cancers-15-02075-t002:** Association of all-cause mortality with CCI in stage I-II NSCLC patients undergoing thoracoscopic resection (N = 61,760).

	No. Death/Person-Years	Mortality Rate (Per 1000 Person-Years) (95% CI)	Age-Adjusted HR (95% CI)	aHR (95% CI) †	aHR (95% CI) §
**CCI**	Overall: 18,833/225,222.3	Overall: 83.6 (82.5–84.8)			
0	8073/117,522.5	68.7 (67.3–70.2)	1 (Reference)	1 (Reference)	1 (Reference)
1	6650/72,935.6	91.2 (89.1–93.3)	1.29 (1.25–1.34)	1.25 (1.21–1.29)	1.24 (1.20–1.28)
2+	4110/34,764.2	118.2 (114.8–121.7)	1.62 (1.56–1.68)	1.55 (1.49–1.61)	1.51 (1.45–1.57)
			*p-trend < 0.01*	*p-trend < 0.01*	*p-trend < 0.01*

Abbreviations: aHR: adjusted hazard ratio; CCI: Charlson comorbidity index; CI: confidence interval. All models were stratified by sex. † Adjusted for age, race/ethnicity, education, income, insurance, and facility type (academic versus non-academic). § Adjusted for all covariates included in the first model, as well as stage at diagnosis, histological type, days from diagnosis to surgery, and receipt of adjuvant chemotherapy.

**Table 3 cancers-15-02075-t003:** Subgroup analysis for the association between CCI and risk of all-cause mortality by selected covariates.

	CCI = 0		CCI = 1		CCI = 2+		
Subgroup	No. Death/Person Years	aHR (95% CI)	No. Death/Person Years	aHR (95% CI)	No. Death/Person Years	aHR (95% CI)	*p*-interaction
**Sex**							
Female (N = 35,148)	3914/73,545.4	1 (Reference)	3127/42,290.4	1.31 (1.25–1.37)	1766/18,122.4	1.65 (1.56–1.75)	
Male (N = 26,612)	4159/43,977.1	1 (Reference)	3523/30,645.1	1.17 (1.12–1.22)	2344/16,641.9	1.40 (1.33–1.47)	
							<0.01
**Age**							
50–64 (N = 18,327)	1821/38,888.4	1 (Reference)	1497/22,383.8	1.26 (1.17–1.35)	754/9070.7	1.43 (1.31–1.56)	
65–74 (N = 26,088)	3082/47,666.8	1 (Reference)	2763/31,758.1	1.27 (1.21–1.34)	1770/15,994.1	1.57 (1.48–1.67)	
75+ (N = 17,345)	3170/30,967.3	1(Reference)	2390/18,793.7	1.17 (1.11–1.24)	1586/9699.4	1.46 (1.38–1.55)	
							0.11
**Days from Diagnosis to Surgery**							
<30 (N = 31,174)	3984/64,292.1	1 (Reference)	3144/37,340.7	1.25 (1.19–1.31)	1890/16,899.0	1.55 (1.47–1.64)	
≥30 (N = 30,586)	4089/53,230.3	1 (Reference)	3506/35,594.8	1.22 (1.16–1.27)	2220/17,865.2	1.47 (1.40–1.55)	
							<0.01
**Facility Type**							
Academic (N = 27,339)	3416/55,558.6	1 (Reference)	2684/30,998.4	1.29 (1.23–1.36)	1673/14,887.5	1.58 (1.49–1.68)	
Non-academic (N = 30,586)	4657/61,963.9	1 (Reference)	3966/41,937.2	1.19 (1.14–1.25)	2437/19,876.8	1.46 (1.39–1.53)	
							<0.01
**Laterality**							
Left (N = 36,374)	3389/47,939.3	1 (Reference)	2728/29,463.2	1.22 (1.16–1.28)	1718/14,295.8	1.50 (1.42–1.59)	
Right (N = 25,131)	4637/69,201.6	1 (Reference)	3885/43,243.3	1.25 (1.20–1.31)	2355/20,350.7	1.52 (1.44–1.60)	
							0.07
**Type of Surgery**							
Sublobar resection (N = 18,007)	2490/377,006.8	1 (Reference)	2309/260,478.5	1.25 (1.18–1.32)	1505/134,750.9	1.50 (1.40–1.60)	
Lobectomy/bilobectomy (N = 42,544)	5249/1010,663.0	1 (Reference)	4127/601,114.5	1.22 (1.17–1.27)	2455/275,842.3	1.49 (1.42–1.57)	
Pneumonectomy (N = 455)	107/10,272.3	1 (Reference)	53/4826.6	1.08 (0.76–1.53)	37/2324.9	1.26 (0.85–1.88)	
Other (N = 754)	227/12,327.0	1 (Reference)	161/8807.2	0.97 (0.79–1.20)	113/4252.6	1.30 (1.03–1.66)	
							<0.01

Abbreviations: aOR: adjusted odds ratio, BMI: body mass index, CI: confidence interval, CCI: Charlson comorbidity index. All models were adjusted for all variables in the full model in Table 2, with the exception of the variable used for stratification.

## Data Availability

The datasets used and/or analyzed during the current study are available from the corresponding author upon reasonable request.

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
