# Peer review of "Survival Differences by Comorbidity Burden among Patients with Stage I/II Non-Small-Cell Lung Cancer after Thoracoscopic Resection"

_cancers, 2023, doi:10.3390/cancers15072075_

Round 1

Reviewer 1 Report

Possibly interesting topic, but no significant advance in current knowledge.

Author Response

- Possibly interesting topic, but no significant advance in current knowledge.

Survival outcomes of early-stage NSCLC depends on many factors. Among these factors, treatment modality is a very important one, but it is not the only factor that can affect survival. Although thoracoscopic resection is a favored option (due to fewer post-procedural complications, shorter length of hospital stay, and faster recovery), it does not guarantee a favorable long-term outcome in early-stage NSCLC patients treated with this technique.

Our study provided a very important message—utilization of thoracoscopic resection may be associated with lower survival rates for early-stage NSCLC if patients have a high comorbidity burden. The positive association between comorbidity burden and mortality indicates that physicians and patients should be aware that burden of co-existing illnesses plays a very essential role in prognosis even though thoracoscopic resection has many advantages compared to traditional open surgery. In addition, our study suggests that impact of comorbidity on mortality is stronger in female NSCLC patients, which is informative and its underlying mechanism has not been well studied in prior research.

Reviewer 2 Report

Dear Authors the topic is interesting as well is clear.

I have only one observation :

2.2. Exposure, outcome, and covariates

“Cancer stage was defined by the 6th or 7th edition of the American 115 Cancers 2023, 15, x FOR PEER REVIEW 4 of 12

Joint Committee of Cancer TNM staging system”.:

WHY YOU DID NOT CONSIDERED THE 8 TH EDITION ?

Author Response

- Dear Authors the topic is interesting as well is clear.

I have only one observation :

2.2. Exposure, outcome, and covariates

“Cancer stage was defined by the 6th or 7th edition of the American 115 Cancers 2023, 15, x FOR PEER REVIEW 4 of 12

Joint Committee of Cancer TNM staging system”.:

WHY YOU DID NOT CONSIDERED THE 8 TH EDITION ?

Thank you for your comment. All patients included in this analysis were diagnosed prior to national implementation of the 8th edition of the AJCC staging manual which started in January 2018. Clarification as to why the 6th and 7th edition were used has been added on Lines 116 & 117.

“Cancer stage was defined by the 6th or 7th edition of the American Joint Committee of Cancer TNM staging system as they were the editions in use during the study period.”

Reviewer 3 Report

I would like to thank Authors for offering me the chance to review this stimulating manuscript and also congratulate them for the interesting results.

Overall, I think that this is a very interesting topic and that the manuscript deserve publication

I have minor comments/advice/questions for the authors:

- results are based on a large number of cases (>60.ooo) and this should be considered and valued as most of the papers on the same subject are referring to a smaller population or are based on multi-institutional series with a limited series of cases. The power of the key message is amplificated.

- I noticed that in the selection process, among 346.999 patients operate for stage I&II NSCLC in the chosen time-window, “only” 80,403 were thoracoscopic resections (61,760 after screening procedure as final sample). This means that almost 77% of cases were open procedures in the selected period. Could this reflect a selection bias? 23% of MIS procedures in 2010-17 is a small number…were these cases selected according to other variables? Maybe academic vs non-academic institutions? Distribution along years was asymmetric: maybe it was around 20% in 2010 and then approaching 30% after 2015 for instance? 23% is a very low threshold for MIS resection….

- table 1, page 6 of 12: 4.4% of non-radical (R1/R2 resection) it is an high number of uncomplete resections according to modern standards. Can the authors supply an explanation for this? Again, maybe academic, or high-volume centers did better than non-academic or low volume ones?

- table 1, page 6 of 12: extent of resection, sub lobar stands for wedge or for anatomical segmentectomies or for both? This deserves a comment as morbidity can be related to a different type of procedures…. I am aware that most databases do not separate wedge vs segments but this should be reported and clearly discussed within the text in my personal opinion as it is relevant.

- a more general comment is that the key message - despite the high numbers and the nice statistics and very well-organized presentation - is overall not so surprising. As most of the impact of comorbidity on short term outcomes has been already and deeply investigated…length of stay, mortality, complications….so intuitively same benefits are expected also in long term…I think this is important for the selection of the topic…

- the explanation of the effect high CCI & reduced overall survival (lines 247 to 264, page 9 out of 12) might probably be improved. I don’t think clinically this could be explained only with cellular and immune system senescence or with chronic inflammation and type of health care delivery. These are very generic despite significant variables. I would personally consider to add in this area some comments on respiratory and cardiac complications as well as infections as major issues leading to mortality. Also the histology and smoking habit should be discussed in respect to the reduced overall survival

Again thank you for asking my comments on this nice paper

Author Response

I would like to thank Authors for offering me the chance to review this stimulating manuscript and also congratulate them for the interesting results.

Overall, I think that this is a very interesting topic and that the manuscript deserve publication

I have minor comments/advice/questions for the authors:

- results are based on a large number of cases (>60.ooo) and this should be considered and valued as most of the papers on the same subject are referring to a smaller population or are based on multi-institutional series with a limited series of cases. The power of the key message is amplificated.

Thank you for your suggestion. We have added that the large sample size is a key strength in our discussion (lines 297-299)

“Furthermore, our sample included over 60,000 NSCLC cases; this large sample size strengthens the precision of our estimates and is significantly larger than many studies of NSCLC patients.”

- I noticed that in the selection process, among 346.999 patients operate for stage I&II NSCLC in the chosen time-window, “only” 80,403 were thoracoscopic resections (61,760 after screening procedure as final sample). This means that almost 77% of cases were open procedures in the selected period. Could this reflect a selection bias? 23% of MIS procedures in 2010-17 is a small number…were these cases selected according to other variables? Maybe academic vs non-academic institutions? Distribution along years was asymmetric: maybe it was around 20% in 2010 and then approaching 30% after 2015 for instance? 23% is a very low threshold for MIS resection….

Thank you for your comment. The 266 thousand patients that were removed for not receiving thoracoscopic resection includes patients who received open surgical resection but also includes patients who did not receive surgical resection at all. We have revised the flowchart to improve clarity.

- table 1, page 6 of 12: 4.4% of non-radical (R1/R2 resection) it is an high number of uncomplete resections according to modern standards. Can the authors supply an explanation for this? Again, maybe academic, or high-volume centers did better than non-academic or low volume ones?

Thank you for your feedback. Upon re-inspection of our dataset, it does appear that non-academic centers are driving the large number of patients who had incomplete resections. We have added details in lines 175-177.

“The majority of patients had complete resection with no residual margins (95.6%). Among the 4.4% who did not receive complete resection, over 60% were treated in non-academic facilities which may have a lower volume of surgical resections.”

- table 1, page 6 of 12: extent of resection, sub lobar stands for wedge or for anatomical segmentectomies or for both? This deserves a comment as morbidity can be related to a different type of procedures…. I am aware that most databases do not separate wedge vs segments but this should be reported and clearly discussed within the text in my personal opinion as it is relevant.

Sublobar resection in this study does include both anatomical segmentectomies and wedge resections. We have conducted an additional sensitivity analysis by sublobar resection type (Supplementary Table 4) and have included a discussion of the results of the sensitivity analysis (lines 163-165 and lines 226-229)

“In our final sensitivity analysis, we explored the relationship between type of sublobar resection (wedge resection, segmental resection, or other), CCI, and mortality.”

“Finally, in our sensitivity analysis among patients who underwent sublobar resection only, effect measures were similar by type of sublobar resection and did not differ substantially from those obtained in the main analysis (Supplementary Table 4).”

- a more general comment is that the key message - despite the high numbers and the nice statistics and very well-organized presentation - is overall not so surprising. As most of the impact of comorbidity on short term outcomes has been already and deeply investigated…length of stay, mortality, complications….so intuitively same benefits are expected also in long term…I think this is important for the selection of the topic…

Thank you for your feedback. Survival of stage I/II NSCLC depends on many factors such as treatment modality, however there are numerous other factors that can affect survival post treatment. While thoracoscopic resection has become the preferred modality for surgical resection of stage I/II NSCLC (due to fewer post-procedural complications, shorter length of hospital stay, and faster recovery), it does not guarantee improved long term survival. The primary message of our study is therefore relevant as it demonstrates that thoracoscopic resection does not necessarily confer a survival advantage among stage I/II NSCLC patients who have a high comorbidity burden. The finding that comorbidity burden is associated with reduced mortality highlights that physicians and patients should be mindful that the presence of comorbidities plays an important role in prognosis even though thoracoscopic resection has many advantages compared to traditional open surgery. Further, the results of our study suggest that the impact of comorbidity burden on mortality is stronger in female NSCLC patients, which is informative and its underlying mechanism warrants further studies.

- the explanation of the effect high CCI & reduced overall survival (lines 247 to 264, page 9 out of 12) might probably be improved. I don’t think clinically this could be explained only with cellular and immune system senescence or with chronic inflammation and type of health care delivery. These are very generic despite significant variables. I would personally consider to add in this area some comments on respiratory and cardiac complications as well as infections as major issues leading to mortality. Also the histology and smoking habit should be discussed in respect to the reduced overall survival

We have added a brief discussion of the potential for post-surgical complications, smoking history, and histological subtype to influence post-surgical survival among patients with a high comorbidity burden (lines 269-278).

“Further, NSCLC patients—particularly those with pre-existing chronic conditions—are  often at risk of cardiac and/or respiratory complications, including but not limited to pneumonia, atrial fibrillation, and venous thromboembolism. Such complications may occur shortly after surgical resection and often cause permanent damage, serving as competing risks for mortality. NSCLC patients with postoperative complications have been found to have reduced 5-year overall survival compared to their counterparts who did not develop postoperative complications. Other factors that can influence prognosis following surgical resection of NSCLC among patients with an increased comorbidity burden include smoking history—which can also influence development of post-surgical complications—and histological type at diagnosis.”

Again thank you for asking my comments on this nice paper

Thank you for your feedback! 

Reviewer 4 Report

Thank you for sending your paper to Cancers. I have learned from reviewing your article.

The thematic you discuss is interesting and still open, thus it could improve our basic knowledge and help to stimulate further research and innovative studies on this important topic. Furthermore, it may help physicians in clinical practice. English is fluent and good. I would recommend your paper to our Colleagues. 

Thank you for this privilege.

Author Response

- Thank you for sending your paper to Cancers. I have learned from reviewing your article.

- The thematic you discuss is interesting and still open, thus it could improve our basic knowledge and help to stimulate further research and innovative studies on this important topic. Furthermore, it may help physicians in clinical practice. English is fluent and good. I would recommend your paper to our Colleagues. 

- Thank you for this privilege.

Thank you for your feedback.

Round 2

Reviewer 1 Report

No further commentary with respect to my first evaluation.